# Severe Acute Kidney Injury in Critically Ill Patients with COVID-19 Admitted to ICU: Incidence, Risk Factors, and Outcomes

**DOI:** 10.3390/jcm10061217

**Published:** 2021-03-15

**Authors:** Muriel Ghosn, Nizar Attallah, Mohamed Badr, Khaled Abdallah, Bruno De Oliveira, Ashraf Nadeem, Yeldho Varghese, Dnyaseshwar Munde, Shameen Salam, Baraa Abduljawad, Khaled Saleh, Hussam Elkambergy, Ali Wahla, Ahmed Taha, Jamil Dibu, Ahmed Bayrlee, Fadi Hamed, Nadeem Rahman, Jihad Mallat

**Affiliations:** 1Medical Sub-Specialties Institute, Department of Nephrology, Cleveland Clinic Abu Dhabi, Abu Dhabi 112412, United Arab Emirates; GhosnM@ClevelandClinicAbuDhabi.ae (M.G.); AttallN@ClevelandClinicAbuDhabi.ae (N.A.); 2Cleveland Clinic Lerner College of Medicine of Case Western Reserve University, Cleveland, OH 44195, USA; 3Critical Care Institute, Cleveland Clinic Abu Dhabi, Abu Dhabi 112412, United Arab Emirates; rashadicu@gmail.com (M.B.); Dr_khaled_Salah@windoslive.com (K.A.); deolivb@clevelandclinicabudhabi.ae (B.D.O.); ashmohnad@hotmail.com (A.N.); VargheY2@ClevelandClinicAbuDhabi.ae (Y.V.); MundeD@ClevelandClinicAbuDhabi.ae (D.M.); SalamS3@ClevelandClinicAbuDhabi.ae (S.S.); AbduljB@ClevelandClinicAbuDhabi.ae (B.A.); salehk@clevelandclinicabudhabi.ae (K.S.); ElkambH@ClevelandClinicAbuDhabi.ae (H.E.); wahlaa@clevelandclinicabudhabi.ae (A.W.); TahaA2@ClevelandClinicAbuDhabi.ae (A.T.); DibuJ@ClevelandClinicAbuDhabi.ae (J.D.); BayrleA@ClevelandClinicAbuDhabi.ae (A.B.); HamedF@ClevelandClinicAbuDhabi.ae (F.H.); RahmanN2@ClevelandClinicAbuDhabi.ae (N.R.); 4Faculty of Medicine, Normandy University, UNICAEN, ED 497, 1400 Caen, France

**Keywords:** acute kidney injury, COVID-19, critically ill, interleukin-6, D-dimer, mechanical ventilation, intensive care unit, outcomes

## Abstract

Background: Critically ill patients with COVID-19 are prone to develop severe acute kidney injury (AKI), defined as KDIGO (Kidney Disease Improving Global Outcomes) stages 2 or 3. However, data are limited in these patients. We aimed to report the incidence, risk factors, and prognostic impact of severe AKI in critically ill patients with COVID-19 admitted to the intensive care unit (ICU) for acute respiratory failure. Methods: A retrospective monocenter study including adult patients with laboratory-confirmed severe acute respiratory syndrome coronavirus-2 (SARS-CoV-2) infection admitted to the ICU for acute respiratory failure. The primary outcome was to identify the incidence and risk factors associated with severe AKI (KDIGO stages 2 or 3). Results: Overall, 110 COVID-19 patients were admitted. Among them, 77 (70%) required invasive mechanical ventilation (IMV), 66 (60%) received vasopressor support, and 9 (8.2%) needed extracorporeal membrane oxygenation (ECMO). Severe AKI occurred in 50 patients (45.4%). In multivariable logistic regression analysis, severe AKI was independently associated with age (odds ratio (OR) = 1.08 (95% CI (confidence interval): 1.03–1.14), *p* = 0.003), IMV (OR = 33.44 (95% CI: 2.20–507.77), *p* = 0.011), creatinine level on admission (OR = 1.04 (95% CI: 1.008–1.065), *p* = 0.012), and ECMO (OR = 11.42 (95% CI: 1.95–66.70), *p* = 0.007). Inflammatory (interleukin-6, C-reactive protein, and ferritin) or thrombotic (D-dimer and fibrinogen) markers were not associated with severe AKI after adjustment for potential confounders. Severe AKI was independently associated with hospital mortality (OR = 29.73 (95% CI: 4.10–215.77), *p* = 0.001) and longer hospital length of stay (subhazard ratio = 0.26 (95% CI: 0.14–0.51), *p* < 0.001). At the time of hospital discharge, 74.1% of patients with severe AKI who were discharged alive from the hospital recovered normal or baseline renal function. Conclusion: Severe AKI was common in critically ill patients with COVID-19 and was not associated with inflammatory or thrombotic markers. Severe AKI was an independent risk factor of hospital mortality and hospital length of stay, and it should be rapidly recognized during SARS-CoV-2 infection.

## 1. Introduction

Since late 2019, severe acute respiratory syndrome coronavirus-2 (SARS-CoV-2) implicated in COVID-19 has infected millions of people all around the world, with hundreds of thousands of deaths [1]. The disease has resulted in a large number of hospitalizations, respiratory failure, and intensive care unit (ICU) admissions. The hospitals in most metropolitan areas have experienced a rapid surge in COVID-19 admissions, with significant mortality [2].

An alarming number of patients also developed acute kidney injury (AKI). Indeed, AKI in patients with COVID-19 infection is reported to range between 6.5% and 46% [2,3,4,5], with the highest ranges in the critically ill (23%–81%) [3,4,5,6,7,8,9,10,11,12]. Differences in the incidence of AKI have resulted from the various definitions of AKI used and the populations studied. Furthermore, AKI has been linked with a higher mortality worldwide [13].

The pathophysiology of AKI related to COVID-19 is incompletely elucidated. Several mechanisms have been suggested to be involved in COVID-19-associated AKI. SARS-CoV-2 can directly injure the tubular cells by binding to the angiotensin-converting enzyme 2 receptors expressed in high amounts in the kidneys’ proximal tubular cells and podocytes [14,15,16,17,18], thus leading to AKI. Severe COVID-19 is also known to cause the excessive release of inflammatory cytokines with high levels of interleukin 6 (IL-6) and maladaptive immune responses [19], thus leading to intrarenal inflammation, increased vascular permeability, and alterations of kidney microcirculation that result in renal hypoperfusion [17,18,20]. Furthermore, a high incidence of acute thrombotic events has been reported in COVID-19 patients [21,22,23]. The presence of fibrin thrombi in the kidney in autopsy findings of patients who died from COVID-19 [24] is in favor of abnormal coagulation that might induce renal microcirculatory dysfunction and AKI.

Data focusing on severe AKI (Kidney Disease Improving Global Outcomes (KDIGO) stages 2 or 3) in ICU are still scarce. Additionally, there have been no reports regarding AKI in patients with COVID-19 admitted to the ICU in the United Arab Emirates (United Arab Emirates). We decided to review our data to analyze the incidence of severe AKI, its predictors, and its association with mortality in critically ill COVID-19 patients in the ICU at Cleveland Clinic Abu Dhabi.

## 2. Methods

This retrospective study was approved by the institutional Ethics Committee of Cleveland Clinic Abu Dhabi on June 8th, 2020 (REC number: A-2020–055) and waived the need for informed consent due to the study’s retrospective nature.

All adult patients (age ≥ 18 years) admitted to our ICU between 1 March and 29 May 2020 with confirmed SARS-CoV-2 infection (virus detected by a real-time reverse-transcriptase–polymerase-chain-reaction assay of a nasopharyngeal sample) and acute respiratory failure requiring high-flow nasal oxygen therapy or mechanical ventilation were included in this study.

### 2.1. Definition of AKI

AKI was defined according to both urinary output and serum creatinine KDIGO criteria [25] as follows: stage 1: increase in serum creatinine by 26.5 µmol.L^−1^ within 48 h, a 1.5–1.9 times baseline value of serum creatinine, or urinary output < 0.5 mL.kg^1^.h^−1^ for 6–12 h within 7 days; stage 2: 2.0–2.9 times baseline value of serum creatinine or urinary output < 0.5 mL.kg^−1^.h^−1^ for ≥ 12 h within 7 days; stage 3: ≥ 3 times baseline value of serum creatinine or to ≥ 353.6 µmol.L^−1^, the initiation of renal replacement therapy (RRT), or urinary output < 0.3 mL.kg^−1^.h^−1^ for ≥24 h or anuria for ≥12 h within 7 days. Baseline creatinine was defined as the best value in the 3 preceding months, or, if unavailable, we used the serum creatinine on the day of hospital admission as the baseline serum creatinine.

### 2.2. Clinical and Laboratory Data

Data on baseline characteristics—including demographics, physiological variables, the presence of medical comorbidities; Sequential Organ Failure Assessment (SOFA) and Simplified Acute Physiology Score (SAPS) II scores; and laboratory values including oxygenation parameters, full blood count, coagulation parameters, and inflammatory markers (C-reactive protein, IL-6, and ferritin)—were collected on admission to the ICU. Data on the use of invasive mechanical ventilation, vasopressors, renal replacement therapy, extracorporeal membrane oxygenation, and other treatments were collected. Ventilatory variables including plateau pressure (Pplat), total positive end-expiratory pressure (PEEP), tidal volume, and driving pressure (Pplat-PEEP) were also captured. The time from symptoms onset to ICU admission was also calculated. Patients were categorized according to whether they developed AKI stages 2 and 3 or no AKI and AKI stage 1 during the ICU stay.

### 2.3. Outcome Measures

The primary outcome was to identify the incidence and risk factors associated with the occurrence of severe AKI (KDIGO stages 2 or 3).

Prespecified secondary outcomes were the associations between severe AKI and all-cause hospital mortality, ICU length of stay, hospital length of stay, and mechanical ventilation duration.

## 3. Statistical Analysis

The normality of data distribution was assessed using the Shapiro–Wilk test and visually checking each variable’s distribution (histogram). Data are expressed as mean ± SD when normally distributed or as median (interquartile range (IQR)) when non-normally distributed. Proportions were used as descriptive statistics for categorical variables. Comparisons of values between independent groups were performed by the 2-tailed Student *t* test or the Mann–Whitney *U* test, as appropriate. The analysis of the discrete data was performed by *χ*2 test or Fisher exact test when the numbers were small. There were missing data (missing at random) for IL-6 (3.6%), ferritin (1.8%), D-dimer (1%), and fibrinogen (8.2%) that were not imputed.

Independent risk factors of AKI stages 2 and 3 were assessed using multivariable logistic regression analysis. Variables associated with AKI stages 2 and 3 (*p* < 0.1) in univariate analysis were included in the model. To respect the rule of 5–9 events per variable in logistic regression [26], we selected the model with less than 10 events per variable that had the lower Akaike information criteria (AIC) and Bayesian information criteria (BIC) values. The potential problem of co-linearity was evaluated using Spearman or Pearson correlation coefficient before running the analysis. The goodness of fit of the model was assessed using Hosmer–Lemeshow test.

Logistic regression analysis was also used to assess whether severe AKI (stages 2 or 3) was independently associated with hospital mortality. As sensitivity analyses, we evaluated if severe AKI was associated with hospital mortality in different models with less than 10 events per variable [26] by keeping age, severity score, and comorbidity variables in each model. Cox proportional hazards regression analysis was also used to assess whether severe AKI was independently associated with time-to-hospital death. Variables associated with hospital mortality (*p* < 0.1) in univariate analysis were included in the Cox model. The proportional hazards assumption was checked based on the scaled Schoenfeld residuals.

The association between severe AKI (stages 2 or 3) and hospital length of stay was evaluated using the competing-risks regression proportional sub hazards analysis based on the method of Fine and Gray. Death before hospital discharge was considered the competing event, and time-to-event analysis was right-censored at hospital discharge. Adjusted competing-risks regression models were fitted to identify whether severe AKI was independently associated with time-to-hospital discharge.

A value of *p* < 0.05 was considered statistically significant, and all reported *p* values are two-sided. Robust standard errors were used to calculate the 95% confidence intervals (CIs) for odds ratios (ORs), hazards ratios (HRs), and subhazard ratios (SHRs). Statistical analyses were performed using Stata/SE 14.2 software for Windows (Stata Corp LLC., College Station, TX, USA).

## 4. Results

### 4.1. Study Population

From 1 March to 29 May 2020, 110 adult patients with acute respiratory failure caused by COVID-19 infection were admitted to the ICU and included in this study (Figure 1). The main characteristics of the cohort are summarized in Table 1. The median age among all patients was 50 years (IQR: 41–59 years), and 98 (89.1%) were men. Among the patients, 65 (59.1%) had at least one comorbidity, 77 (70.0%) received invasive mechanical ventilation, and 66 (60.0%) required vasopressor support. The median time from symptoms onset to ICU admission was 5 days (IQR: 3–7 days).

Fifty patients (45.4%) developed severe AKI (stage 2 or 3), and 60 (54.5%) had no AKI (46.4%) or AKI stage 1 (8.2%); see Table 1. The median time from ICU admission to AKI occurrence was 0 days (IQR: 0–7 days). Among patients who developed severe AKI (stages 2 or 3), 27 (54%) required renal replacement therapy. The median time from ICU admission to renal replacement therapy was 2 days (IQR: 0–9 days).

Patients in the severe AKI group were older and had a higher rate of comorbidities, higher severity scores, and lower a mean arterial pressure on ICU admission than the other group (Table 1). Regarding laboratory data on ICU admission, only leucocyte, D-dimer, and creatinine levels were significantly higher in the severe AKI group than in the other group (Table 1). IL-6, C-reactive protein, ferritin, and fibrinogen levels did not significantly differ between the two groups (Table 1). Urine output and fluid balance were not significantly different between the two groups. The treatments received in the ICU (invasive mechanical ventilation, vasopressors, diuretics, and extracorporeal membrane oxygenation) were significantly higher in the severe AKI group (Table 1). The other treatments did not differ significantly between the two groups.

### 4.2. Primary Outcome

In univariate analysis (without adjustment), age, comorbidities (diabetes mellitus, hypertension, and chronic kidney disease), severity scores (SOFA and SAPS II), mean arterial pressure, leucocyte count, D-dimer, creatinine on admission, vasopressor use, invasive mechanical ventilation, and extracorporeal membrane oxygenation were found to be associated (*p* < 0.1) with severe AKI (Table 1). In the multivariable logistic regression analysis, after including the variables mentioned above, age, creatinine level on ICU admission, the use of invasive mechanical ventilation, and extracorporeal membrane oxygenation were independently associated with the development of severe AKI stages (Appendix A). Instead of SAPS II, SOFA score was included in the model because of the collinearity between the two variables (r = 0.73, *p* < 0.001). However, including SAPS II instead of the SOFA score resulted in the same findings (results not shown). Table 2 shows the logistic regression model with the lowest AKI and BIC values, including less than nine events per variable. The same variables (age, admission creatinine, invasive mechanical ventilation, and extracorporeal membrane oxygenation) were found as risk factors for severe AKI development.

### 4.3. Secondary Outcomes

Overall, 27 patients (24.5%) died in the hospital. Among them, 23 patients (46%) were in the severe AKI group, and only 4 (6.7%) were in the other group (*p* < 0.001). In univariate analysis, age, comorbidities (hypertension), SAPS II, leucocyte count, platelet count, ferritin, PaO_2_/FiO_2_ ratio, lactate levels, severe AKI, vasopressor use, invasive mechanical ventilation, extracorporeal membrane oxygenation, and hydroxychloroquine were found to be associated (*p* < 0.1) with hospital mortality (Table 3). In the multivariable logistic regression analysis, after adjusting for the above confounder variables, severe AKI was found to be independently associated with a higher hospital mortality (OR = 29.7 (95% CI: 4.10–215.8), *p* = 0.001); see Table 4. After a sensitivity analysis, we found that severe AKI was still independently associated with hospital mortality in the different models that included less than 10 events per variable (Appendix A). We did not adjust *p*-values for multiple comparisons because they were considered to be exploratory analyses. Severe AKI was also associated with time-to-hospital death in univariate analysis (HR = 4.25 (95% CI: 1.47–12.30), *p* = 0.008). However, after adjusting for confounding variables, severe AKI was no longer associated with time-to-hospital death (HR = 4.00 (95% CI: 0.80–20.00), *p* = 0.09); see Table 5.

Hospital length of stay (31 days (IQR: 19–51 days) vs. 19 days (IQR: 12–28 days), *p* = 0.002), ICU length of stay (21 days (IQR: 4–35 days) vs. 9 days (IQR: 6–17 days), *p* < 0.001), and duration of mechanical ventilation (21 days (IQR: 10–37 days) vs. 11 days (IQR: 8–15 days), *p* = 0.002) were significantly longer in the severe AKI group than in the other group. In the multivariable competing-risks regression analysis, after adjusting for confounding variables, severe AKI was independently associated with a longer hospital length of stay (SHR = 0.26 (95% CI: 0.14–0.51), *p* < 0.001); see Figure 2 and Appendix A.

Among the 50 patients with severe AKI, 27 (54%) were discharged alive from the hospital. At hospital discharge, 20 of 27 (74.1%) recovered normal or baseline renal function.

## 5. Discussion

In this study of critically ill patients with COVID-19 and acute respiratory failure, the incidence of severe AKI was 45.4%. Age, creatinine level on admission, invasive mechanical ventilation, and extracorporeal membrane oxygenation were associated with severe AKI development. Furthermore, severe AKI was associated with a higher hospital mortality rate and a longer hospital length of stay.

The reported incidence of overall AKI using KDIGO definition in critically ill patients admitted to the ICU ranged from 50% to 81% [3,4,8,10,11]. In a retrospective study of 100 COVID-19 patients admitted to the ICU, Joseph et al. found a 37% incidence of severe AKI (stages 2 and 3) [8]. Other studies showed severe AKI incidence ranging from 40.6% to 57.4% in COVID-19 ICU patients [3,4,10,11]. We found a 45.5% incidence of severe AKI. The differences in the reported incidence rates of severe AKI might be explained by differences in severity of COVID-19 patients and the different methods/definitions used to estimate missing baseline creatinine, which can lead to disparities in the incidence of AKI of up to 15% [27]. Additionally, our incidence rate of patients requiring renal replacement therapy was higher (24.5%) than that reported in the studies of Joseph et al., Doher et al., and Fominskiy et al. (13%, 16.9%, and 17.7%, respectively) [8,10,11], but it was in line with other reports ranging between 20.4% and 31% [3,4,12,28].

The risk factors of developing AKI vary between studies [3,4,8,10]. Hirsch et al. found that older age, the presence of comorbidities (hypertension, diabetes, and cardiovascular disease), and the need for invasive mechanical ventilation and vasopressor medications were independently associated with AKI in a large population of patients hospitalized with COVID-19 (*n* = 5449) in metropolitan New York [4]. In a retrospective study of patients admitted to the hospital with COVID-19, Chan et al. observed that chronic kidney disease, male gender, and admission potassium were independent predictors of severe AKI (stage 3) [3]. In ICU patients with COVID-19, only chronic kidney disease and modified SOFA score on admission were independently associated with AKI occurrence [8]. In critically ill patients with COVID-19 admitted to the ICU, the use of diuretics, invasive mechanical ventilation, and creatinine level on admission were independently associated with AKI [10]. In our study, the use of invasive mechanical ventilation, age, creatinine level on admission, and extracorporeal membrane oxygenation were associated with the development of severe AKI (stages 2 and 3). The most common predictors of AKI across all those studies were older age, comorbidities (specifically hypertension and chronic kidney disease), and the need for invasive mechanical ventilation.

COVID-19 has been characterized by elevated cytokine levels (cytokine storm) and a hyperinflammatory state induced by SARS-CoV-2 infection [18,29]. High levels of IL-6 were found to be associated with the development of severe COVID-19 infection and mortality [30,31]. It can be assumed that IL-6 may contribute to AKI in COVID-19 patients by inducing endothelial and tubular dysfunction. Indeed, the harmful effect of IL-6 has been shown in different models of AKI, including ischemic and sepsis-induced AKI [31,32,33]. In our study, IL-6, C-reactive protein, and ferritin were not independently associated with severe AKI (Table 2). Our findings were in line with those of Joseph et al., who found no association between IL-6, ferritin, and AKI development [8]. The invoked “cytokine storm” induced by COVID-19 infection has been recently challenged. It has been shown that even if blood IL-6 levels were elevated in patients with COVID-19, they were lower than the values typically reported in acute respiratory distress syndrome [34]. These findings question the premise of whether the degree of inflammation in COVID-19 infection is more extensive than the systemic inflammatory response seen in other infection-associated critical illness.

A hypercoagulable state and a high incidence of acute thrombotic events have been frequently reported in patients with COVID-19 [21,22,23,35]. The presence of thrombi in glomerular loops suggested that coagulation dysregulation can participate in AKI development [24]. However, the absence of association between D-dimer, fibrinogen levels, and AKI in our study and previous reports [8] does not support the evidence of a role of hypercoagulability in COVID-19-induced AKI. Until now, there has been no evidence suggesting that the pathophysiological mechanisms of AKI in COVID-19 are different from sepsis-associated AKI [36]. Indeed, acute tubular necrosis was the predominant pathologic finding in autopsies and kidney biopsies of patients with COVID 19 [37,38]. Until more is known about the pathophysiology of COVID-related AKI, the treatment will continue to be supportive, including careful fluid and hemodynamic management, the avoidance of potentially nephrotoxic agents, and the individualized application of renal replacement therapy [36,39].

Similar to our results, Chan et al. [3] found that AKI was independently associated with death in ICU patients with COVID-19 (adjusted OR = 11.4 (95% CI: 7.2–18)). Additionally, in 201 critically ill COVID-19 patients, another study showed that AKI was significantly related to hospital mortality [10]. On the contrary, Forest et al. [26] did not observe a significant association between renal replacement therapy and death (adjusted OR = 2.3 (95% CI: 0.98–5.5)) in 330 critically ill COVID-19 patients admitted to the ICU. Contrary to previous reports [8,10], we did not find an independent association between severe AKI and an instantaneous risk of death (HR = 4.00 (95% CI: 0.80- 20.00); see Table 5)—only with the likelihood of death over the hospital stay (Table 4). Fominskiy et al. did not also find a relationship between AKI and instantaneous risk of death (adjusted HR = 1.32 (95% CI: 0.48–3.63)) in 96 mechanically ventilated patients with COVID-19 [11]. In line with our results, Doher et al. [10] found that AKI was independently associated with longer hospital length of stay (adjusted SHR = 0.31 (95% CI: 0.22–0.44)). Therefore, across most of these studies, AKI in COVID-19 patients seems to be associated with adverse outcomes, and its awareness must be highlighted.

We observed a higher rate of AKI recovery (74.1%) than those reported by Chan et al. [3] and Pei et al. [40] (which were 65% and 45.7%, respectively) in hospitalized COVID-19 patients, even though our population was sicker than those in the two studies. This might have been because our patients were younger (median age: 50 years) compared to the studies of Chan et al. [3] (median age: 64 years) and Pei et al. [40] (mean age: 63 years). Differences in management might also account for the different rates of renal recovery between studies.

Our study had certain strengths. To the best of our knowledge, this is the first report on AKI in critically ill patients from United Arab Emirates. We focused on severe AKI (stages 2 and 3) in critically ill COVID-19 patients with acute respiratory failure, of which a large proportion was managed with invasive mechanical ventilation (70%) and vasopressors (60%). This contrasts with many other published studies that looked at all comers without a specific emphasis on critically ill patients. Indeed, data on severe AKI in these patients are scarce. Additionally, we had a complete follow-up on all our patients. All patients were discharged from the hospital or died before the conception of the study. Furthermore, we reported data on renal recovery in these patients.

Our study had some limitations. First, it was a retrospective study with the potential for selection bias. However, we included a large number of confounders, and after careful adjustment, we were able to determine the potential risk factors of severe AKI and to demonstrate that severe AKI was an independent predictor of hospital mortality. Nevertheless, our data did not include the use of nephrotoxin agents such as aminoglycosides, which are well known as potential risk factors of AKI development. Second, it was a single-center study performed in a specific region of the world, so the results might not be generalized to other centers. Third, in many cases (60%), we did not have a baseline creatinine value, and using the admission value may have overestimated the incidence of AKI. However, most of our patients were young, and only 5.4% had chronic kidney disease. Thus, we think that the baseline creatinine value was normal in the majority of our patients, and the high creatinine levels observed in some patients on hospital admission truly reflected the presence of AKI.

## 6. Conclusions

In critically ill patients with COVID-19 and acute respiratory failure, we found that the occurrence of severe AKI was common and not associated with inflammatory (IL-6, ferritin, and C-reactive protein) or thromboembolism (D-dimer and fibrinogen) markers. Severe AKI was an independent risk factor of hospital mortality and hospital length of stay, and it should be recognized rapidly during SARS-CoV-2 infection. Prospective studies are needed to confirm our results.

## Figures and Tables

**Figure 1 jcm-10-01217-f001:**
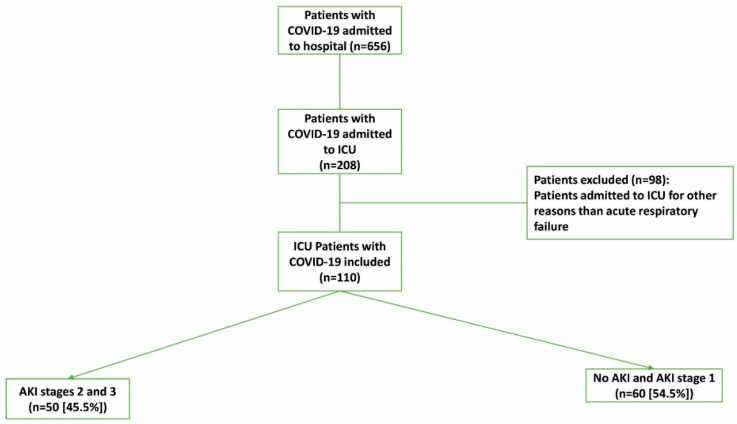
Flow chart of COVID-19 patients admitted to the intensive care unit (ICU). AKI: acute kidney injury.

**Figure 2 jcm-10-01217-f002:**
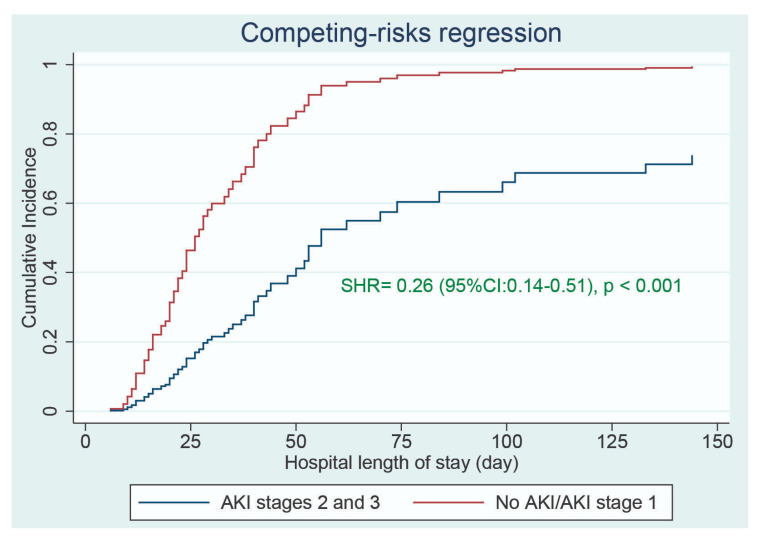
Comparisons of cumulative incidence function of hospital discharge between patients who had severe AKI (KDIGO stages 2 and 3) and those who did not. SHR: subhazard ratio.

**Table 1 jcm-10-01217-t001:** Comparisons of baseline characteristics, laboratory data, and treatments during ICU stay between severe AKI (stage 2 and 3) and AKI stage 1 and no-AKI groups.

Variables	All Patients (*n* = 110)	Severe AKI (*n* = 50)	No-AKI and AKI Stage 1 (*n* = 60)	*p*-Value
**Age, year**	50 (40–59]	54 (45–63)	42 (38–54)	<0.001
**Male, *n* (%)**	98 (89.1)	47 (94.0)	51 (85.0)	1.00
**Body mass index, kg·m^−2^**	26.2 (23.8–30.1)	27.0 (23.8–31.0)	26.0 (23.7–29.4)	0.67
**Obesity (BMI ≥ 30 kg·m^−2^)**	36 (32.7)	19 (38.0)	17 (28.3)	0.28
**SOFA score**	5.0 (3.0–8.0)	7.4 (4.0–11.0)	4.0 (3.0–7.0)	0.001
**SAPS II score**	32 (24–45)	36 (27–51)	29 (21–39)	0.002
**Patients with at least one comorbidity, *n* (%)**	65 (59.1)	35 (70.0)	30 (50.0)	0.034
**Comorbidities distribution, *n* (%)**				
**Diabetes mellitus**	42 (38.2)	27 (54.0)	15 (25.0)	0.002
**Hypertension**	39 (35.4)	25 (50.0)	14 (23.3)	0.004
**Chronic artery disease**	9 (8.3)	6 (12.2)	3 (5.0)	0.29
**Chronic kidney disease**	6 (5.4)	5 (10.0)	1 (1.7)	0.09
**Time from symptoms to ICU admission, day**	5 (3–7)	5 (4–7)	5 (3–7)	0.61
**Vital signs on ICU admission**				
**Temperature (max) ≥38 °C, *n* (%)**	45 (40.9)	20 (40.0)	25 (41.7)	0.86
**Heart rate (max), beats·min^−1^**	105 ± 19	104 ± 19	105 ± 19	0.92
**Lowest mean arterial pressure, mmHg**	70 (65–80)	68 (63–75)	71 (66–82)	0.055
**Laboratory data on ICU admission**				
**C-reactive protein, mg·L^−1^**	138 (63–225)	166 (83–249)	118 (50–220)	0.18
**Leucocyte count, × 10^9^ L^−1^**	8.9 (6.2–12.0)	9.7 (6.5–13.1)	8.2 (5.7–11.7)	0.09
**Lymphocyte count, × 10^9^ L^−1^**	0.78 (0.49–1.09)	0.74 (0.46–1.05)	0.80 (0.54–1.15)	0.38
**Lymphocytes ≤ 1 × 10^9^ L^−1^; *n* (%)**	80 (72.7)	37 (74.0)	43 (71.7)	0.78
**Platelet count, × 10^9^ L^−1^**	233 (164–301)	238 (160–285)	226 (178–309)	0.66
**D-dimer, µg·mL^−1^ (normal reference: <0.05)**	2.6 (0.9–4.0)	3.8 (2.2–4.0)	1.4 (0.8–4.0)	0.003
**D-dimer ≥ 2 µg·mL^−1^, *n* (%)**	65 (59.1%)	38 (76.0)	27 (45.0)	0.001
**Fibrinogen, g·L^−1^**	6.2 (5.0–7.2)	6.0 (5.0–7.0)	6.4 (5.0–7.2)	0.80
**Ferritin, µg·L^−1^ (reference range: 36–480)**	1515 (809–2474)	1644 (844–2582)	1454 (788–2303)	0.32
**Interleukin 6, ng·L^−1^**	219 (106–839)	179 (128–1516)	260 (89–677)	0.57
**Total bilirubin, µmol·L^−1^**	10.7 (7.7–16.4)	10.9 (8.0–20.5)	10.3 (7.0–15.1)	0.32
**Creatinine, µmol·L^−1^**	77 (62–110)	123 (72–303)	68 (57–79)	<0.001
**KDIGO stage, *n* (%)**				
**No AKI**	51 (46.4)	0 (0)	51 (85.0)	
**Stage 1**	9 (8.2)	0 (0)	9 (15.0)	
**Stage 2**	10 (9.1)	10 (20.0)	0 (0)	
**Stage 3**	40 (36.4)	40 (80.0)	0 (0)	
**Urine output day 1, mL·kg^−1^·hr^−1^ (*n* = 106)**	0.53 (0.24–0.91)	0.47 (0.17–0.94)	0.55 (0.39–0.88)	0.33
**Urine output day 2, mL·kg^−1^·hr^−1^ (*n* = 107)**	0.79 (0.40–1.19)	0.73 (0.37–1.14)	0.83 (0.43–1.37)	0.15
**Urine output day 3, mL·kg^−1^·hr^−1^ (*n* = 108)**	0.86 (0.39–1.33)	0.86 (0.31–1.28)	0.86 (0.48–1.37)	0.64
**Cumulative fluid balance day 3, mL**	1027 (−384–2826)	1202 (−247–2464)	540 (−835–3211)	0.71
**Cumulative fluid balance day 7, mL**	995 (−754–4134)	860 (−907–4137)	1225 (−3.5–3600)	0.50
**Treatments during ICU stay, *n* (%)**				
**Vasopressor support**	66 (60)	42 (84.0)	24 (40.0)	<0.001
**Renal replacement therapy**	27 (24.5)	27 (54.0)	0 (0)	<0.001
**Invasive mechanical ventilation**	77 (70.0)	45 (90.0)	32 (53.3)	<0.001
**Extracorporeal membrane oxygenation**	9 (8.2)	7 (14.0)	2 (3.3)	0.08
**Tocilizumab**	96 (87.3)	43 (86.0)	53 (88.3)	0.78
**Methylprednisolone**	42 (38.5)	19 (38.8)	23 (38.3)	0.96
**Hydroxychloroquine**	45 (40.9)	18 (36.0)	27 (45.0)	0.34
**Favipiravir**	30 (27.3)	13 (26.0)	17 (28.3)	0.78
**Lopinavir/ritonavir**	30 (27.3)	14 (28.0)	16 (26.7)	0.88
**Convalescent plasma**	29 (26.4)	14 (28.0)	15 (25.0)	0.72
**Diuretics**	88 (80)	41 (82)	47 (78.3)	0.63

SOFA, Sequential Organ Failure Assessment; SAPS, Simplified Acute Physiology Score; ICU, intensive care unit; AKI, Acute kidney injury unit; KDIGO, Kidney Disease Improving Global Outcomes. Data are shown as mean ± SD, median (1st–3rd quartile), and count (%). *p* ≤ 0.05 was considered statistically significant. Severe AKI is defined as KDIGO stages 2 and 3.

**Table 2 jcm-10-01217-t002:** Factors associated with severe AKI (stages 2 and 3) in multivariable logistic regression analysis.

Variables	Odds Ratio	95% Confidence Interval	*p*-Value
**Age, year**	1.08	1.03–1.14	0.003
**SOFA score**	0.99	0.78–1.25	0.93
**Comorbidities, (reference: no)**	1.05	0.25–4.43	0.94
**Lowest mean arterial pressure, mmHg**	1.00	0.94–1.06	0.97
**Invasive mechanical ventilation, (reference: no)**	33.44	2.20–507.77	0.011
**Creatinine at ICU admission,** **µmol·L^−1^**	1.04	1.008–1.065	0.012
**Vasopressor support, (reference: no)**	0.97	0.16–6.05	0.98
**Extracorporeal membrane oxygenation, (refer: no)**	11.42	1.95–66.70	0.007

SOFA, Sequential Organ Failure Assessment; ICU, intensive care unit. *p* ≤ 0.05 was considered statistically significant. Goodness-of-fit test: *p* = 0.98. Akaike information criteria (AIC): 84.56486. Bayesian information criteria (BIC): 108.6203

**Table 3 jcm-10-01217-t003:** Comparisons of baseline characteristics, laboratory data, and treatments during ICU stay between survival and death groups.

Variables	Death (*n* = 27)	Survival (*n* = 83)	*p*-Value
**Age, year**	53 (45–67)	46 (39–57)	0.03
**Male, *n* (%)**	24 (89.9)	74 (89.2)	1.00
**Body mass index, kg·m^−2^**	26.7 (23.3–31.2)	26.2 (24.0–29.4)	0.74
**Obesity (BMI ≥ 30 kg·m^−2^)**	12 (44.4)	24 (28.2)	0.13
**SOFA score**	7.0 (3.0–11.0)	5.0 (3.0–8.0)	0.12
**SAPS II score**	40 (27–53)	31 (24–42)	0.04
**Patients with at least one comorbidity, *n* (%)**	20 (74.1)	45 (54.2)	0.07
**Comorbidities distribution, *n* (%)**			
**Diabetes mellitus**	11 (40.7)	31 (37.5)	0.75
**Hypertension**	14 (51.8)	25 (30.1)	0.04
**Coronary artery disease**	4 (14.8)	5 (6.1)	0.22
**Chronic kidney disease**	2 (7.4)	4 (4.8)	0.63
**Time from symptoms to ICU admission, day**	5 (4–7)	5 (3–7)	0.71
**Vital signs on ICU admission**			
**Temperature (max) ≥38 °C, *n* (%)**	8 (29.6)	37 (44.6)	0.17
**Heart rate (max), beats·min^−1^**	105 ± 20	104 ± 19	0.89
**Lowest mean arterial pressure, mmHg**	70 (61–75)	71 (65–80)	0.24
**Laboratory data on ICU admission**			
**C-reactive protein, mg·L^−1^**	110 (50–194)	138 [63–229)	0.47
**Leucocyte count, × 10^9^ L^−1^**	10.2 (6.2–16.0)	8.7 (6.2–11.5)	0.07
**Lymphocyte count, × 10^9^ L^−1^**	0.81 (0.47–1.09)	0.78 (0.52–1.12)	0.90
**Lymphocytes ≤ 1 × 10^9^ L^−1^; *n* (%)**	19 (70.4)	61 (73.5)	0.78
**Procalcitonin, ng·L^−1^**	0.50 (0.21–1.91)	0.39 (0.17–3.29)	0.95
**Platelet count, × 10^9^ L^−1^**	185 (144–274)	240 (184–311)	0.054
**D-dimer, µg·mL^−1^ (normal reference: <0.05)**	3.3 (0.9–4.0)	2.6 (0.9–4.0)	0.25
**D-dimer ≥ 2 µg·mL^−1^, *n* (%)**	17 (63.0)	48 (57.8)	0.64
**Fibrinogen, g·L^−1^ (*n* = 95)**	5.6 (4.6–6.6)	6.3 (5.0–7.2)	0.17
**Ferritin, µg·L^−1^ (reference range: 36–480)**	2015 (1260–3020)	1367 (769–2268)	0.06
**Interleukin 6, ng·L^−1^**	173 (127–3046)	252 (86–822)	0.82
**Total bilirubin, µmol·L^−1^**	10.7 (8.0–16.7)	10.7 (7.3–16.0)	0.55
**Creatinine, µmol·L^−1^**	72 (65–172)	77 (61–104)	0.73
**PaO_2_/FiO_2_, mmHg**	67 (52–89)	98 (70–159)	0.002
**PaCO_2_, mmHg**	45 (32–65)	39 (32–50)	0.28
**Lactate, mmol·L^−1^**	1.7 (1.4–2.1)	1.30 (1.2–1.6)	<0.001
**Severe AKI (stages 2 and 3), *n* (%)**	23 (85.2)	27 (35.2)	<0.001
**Cumulative fluid balance day 3, mL**	1202 (−321–3050)	895 (−770–2826)	0.44
**Cumulative fluid balance day 7, mL**	1869 (−500–4348)	981 (−797–3816)	0.62
**Treatments during ICU stay, *n* (%)**			
**Vasopressor support**	24 (88.9)	42 (50.6)	<0.001
**Renal replacement therapy**	13 (48.1)	14 (16.7)	0.001
**Invasive mechanical ventilation**	23 (85.2)	54 (65.1)	0.055
**Extracorporeal membrane oxygenation**	5 (18.5)	4 (4.8)	0.038
**Tocilizumab**	22 (81.5)	74 (89.2)	0.33
**Methylprednisolone**	11 (40.7)	31 (37.8)	0.79
**Hydroxychloroquine**	7 (25.9)	38 (45.8)	0.07
**Favipiravir**	6 (22.2)	24 (28.9)	0.62
**Lopinavir/ritonavir**	5 (18.2)	25 (30.2)	0.32
**Convalescent plasma**	10 (37.0)	19 (22.9)	0.15
**Tidal volume, mL·kg^−1^ IBW**	6.6 (5.4–7.5)	6.5 (5.3–7.1)	0.78
**PEEP, cmH_2_O**	12 (10–14)	12 (10–14)	0.33
**Plateau pressure, cmH_2_O**	28 (27–30)	28 (26–30)	0.43
**Driving pressure, cmH_2_O**	16 (14–19)	16 (13–19)	0.99

SOFA, Sequential Organ Failure Assessment; SAPS, Simplified Acute Physiology Score; ICU, intensive care unit; AKI, acute kidney injury unit; PEEP, positive end-expiratory pressure; IBW, ideal body weight; PaO_2_, arterial oxygen pressure; PaCO_2_, arterial carbon dioxide pressure; FiO_2_, inspiratory oxygen fraction. Data are shown as mean ± SD, median (1st–3rd quartile), and count (%). *p* ≤ 0.05 was considered statistically significant.

**Table 4 jcm-10-01217-t004:** Factors associated with hospital mortality (multivariable logistic regression analysis).

Variables	Odds ratio	95% Confidence Interval	*p*-Value
**Severe AKI (stages 2 and 3)**	29.73	4.10–215.77	0.001
**Age, year**	1.02	0.93–1.12	0.59
**SAPS II**	0.96	0.89–1.04	0.32
**Comorbidities**	6.12	0.73–51.32	0.09
**Platelet count, ×** **10^9^ L^−1^**	0.98	0.97–0.99	0.024
**Invasive mechanical ventilation**	0.12	0.004–3.063	0.20
**Ferritin, µg·L^−1^**	1.00	1.00–1.00	0.09
**Interleukin 6, ng·L^−1^**	1.00	1.00–1.00	0.16
**PaO_2_/FiO_2_, mmHg**	0.99	0.97–1.00	0.09
**Lactate, mmol·L^−1^**	1.50	0.51–4.38	0.46
**Vasopressor support, (reference: no)**	77.87	1.25–4861.70	0.039
**Leucocyte count, ×** **10^9^ L^−1^**	1.07	0.90–1.27	0.46
**Extracorporeal membrane oxygenation, (refer: no)**	56.90	1.17–2771.79	0.042
**Hydroxychloroquine**	0.26	0.01–5.23	0.38

SAPS, Simplified Acute Physiology Score; PaO_2_, arterial oxygen pressure; FiO_2_, inspiratory oxygen fraction. *p* ≤ 0.05 was considered statistically significant. Goodness-of-fit test: *p* = 0.99.

**Table 5 jcm-10-01217-t005:** Factors associated with hospital mortality (multivariable Cox regression analysis).

Variables	Hazards Ratio	95% Confidence Interval	*p*-Value
**Severe AKI (stages 2 and 3), (reference: no AKI/AKI stage 1)**	4.00	0.80–20.00	0.092
**Age, year**	1.00	0.95–1.05	0.99
**SAPS II**	0.98	0.93–1.04	0.55
**Comorbidities**	2.60	0.67–10.10	0.17
**Platelet count, ×** **10^9^ L^−1^**	1.00	0.98–1.00	0.45
**Invasive mechanical ventilation**	0.09	0.005–1.49	0.09
**Ferritin, µg·L^−1^**	1.00	1.00–1.00	0.02
**Interleukin 6, ng·L^−1^**	1.00	1.00–1.00	0.18
**PaO_2_/FiO_2_, mmHg**	0.99	0.98–1.00	0.61
**Lactate, mmol·L^−1^**	1.59	1.008–2.507	0.046
**Vasopressor support, (reference: no)**	20.38	1.22–339.65	0.036
**Leucocyte count, ×** **10^9^ L^−1^**	1.00	0.92–1.08	0.99
**Extracorporeal membrane oxygenation, (refer: no)**	9.90	1.59–61.70	0.014

SAPS, Simplified Acute Physiology Score; AKI, acute kidney injury; PaO_2_, arterial oxygen pressure; FiO_2_, inspiratory oxygen fraction. *p* ≤ 0.05 was considered statistically significant. Goodness-of-fit test: *p* = 0.34.

## Data Availability

The data presented in this study are available on request from the corresponding author. The data are not publicly available due to the Ethics Committee restrictions.

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
