# Peer review of "Severe Acute Kidney Injury in Critically Ill Patients with COVID-19 Admitted to ICU: Incidence, Risk Factors, and Outcomes"

_jcm, 2021, doi:10.3390/jcm10061217_

Round 1
Reviewer 1 Report
The authors relate their experience with COVID associated AKI in critically ill patients
MINOR
- The independent risk factors for AKI studied by the authors did not include nephrotoxins such as aminoglycosides. This limitation should be included in the discussion section
- How do the authors explain the extreme predominance of males admitted to the ICU. Was there gender bias in the selection of those critically ill patients admitted to the ICU or were men so much more prone to a worse course
- pg 10 line 268, change men's to male
- The authors might consider acknowledging that admission to the ICU is a less than ideal surrogate to separate critically ill COVID patients from those who are not critically ill. Many hospitals had their ICUs filled to capacity and many critically ill COVID patients were treated in non-ICU spaces that functioned as makeshift intensive care units. this information is not relevant to this study but is relevant to interpretation of the literature on COVID AKI in critically ill patients
Author Response
Reviewer#1
Reviewer: Minor. 1) The independent risk factors for AKI studied by the authors did not include nephrotoxins such as aminoglycosides. This limitation should be included in the discussion section
Response: We thank the reviewer for this important comment. We totally agree with the reviewer. We have now added in the limitations of the study lines 340-342 the following:” Nevertheless, our data did not include the use of nephrotoxin agents such as aminoglycosides, which are well known as potential risk factors of AKI development.” We hope that we have addressed the reviewer’s concern appropriately.
Reviewer: 2) How do the authors explain the extreme predominance of males admitted to the ICU. Was there gender bias in the selection of those critically ill patients admitted to the ICU, or were men so much more prone to a worse course.
Response: We thank the reviewer for this important point. In our hospital, we received many expatriates’ workers from India, Pakistan, the Philippines, etc. that were predominantly male. However, we do not think that a very high proportion of male patients (89.1%) would have impacted the results. Indeed, the distribution of male gender was well balanced between the severe AKI and non-AKI groups (94% vs. 85%, p=1.00) and between survivors and non-survivors groups (89.2% vs. 89.9%, p=1.00). We hope that we have addressed the reviewer’s concern appropriately.
Reviewer: 3) pg 10 line 268, change men's to male
Response: We thank the reviewer for the thorough reading of the manuscript. We have now changed men’s to male as suggested by the reviewer (line 268).
Reviewer: 4) The authors might consider acknowledging that admission to the ICU is a less than ideal surrogate to separate critically ill COVID patients from those who are not critically ill. Many hospitals had their ICUs filled to capacity and many critically ill COVID patients were treated in non-ICU spaces that functioned as makeshift intensive care units. this information is not relevant to this study but is relevant to interpretation of the literature on COVID AKI in critically ill patients
Response: We thank the reviewer for this important point, and we cannot agree more with his/her comment. Our results apply to critically ill patients with COVID-19 if they are admitted to the ICU or to any unit transformed to admit critically ill patients in the hospital, of course. However, it is not evident to integrate this information in the discussion as it is not relevant to this study, as rightly stated by the reviewer. Nevertheless, to try to clarify this issue, we changed the sentence in the "conclusion" (line 351):" In critically ill patients with COVID-19 admitted to ICU for acute respiratory failure, etc." to "In critically ill patients with COVID-19 and acute respiratory failure, etc." We hope that we have addressed the reviewer's concern appropriately.
We thank the reviewer for his/her comments that have helped improve the manuscript's quality.
Reviewer 2 Report
Thank you for the opportunity to review this work. Authors conducted this retrospective observational study to find the risk factors of Covid-19 associated AKI and its influence on mortality. Authors found that age, IMV, creatinine on admission, and ECMO were associated with AKI in this population. AKI was also associated with hospital mortality. These results are similar with previous studies with larger sample size than this study, which hamper the significance of this study. However, this study still has some significance because of the rigorous definition of AKI (KDIGO criteria) and complete follow-up.
I think interpretation and conclusion of the results are generally appropriate. But, I'd like to point out some minor comments.
Comments to the authors:
- About half of the patients was excluded from this study because they did not have acute respiratory failure. However, I consider that the main reason of ICU admission in patients with Covid-19 should be respiratory failure, because SARS-CoV-2 infection would mainly affect the lungs. What it the definition of the respiratory failure in your inclusion criteria? And, what is the reason of ICU admission of excluded patients? I think these are important to interpret the influence of selection bias on your results.
- How many patients missed the premorbid creatinine value? In some patients, the creatinine on the day of hospital admission might be higher than the premorbid one because of the AKI. If my understanding is correct, your study underestimated the AKI prevalence, because you applied the higher baseline creatinine value in some patients. Please confirm the correctness of your following sentence in discussion section.
Page 11, Line 342-344
"Third, in many cases, we did not have a baseline creatinine value, and using the admission value may have overestimated the incidence of AKI."
Author Response
Reviewer#2
Reviewer: However, this study still has some significance because of the rigorous definition of AKI (KDIGO criteria) and complete follow-up. I think interpretation and conclusion of the results are generally appropriate.
Response: We thank the reviewer for the supportive comment.
Reviewer: 1) About half of the patients was excluded from this study because they did not have acute respiratory failure. However, I consider that the main reason of ICU admission in patients with Covid-19 should be respiratory failure, because SARS-CoV-2 infection would mainly affect the lungs. What it the definition of the respiratory failure in your inclusion criteria? And, what is the reason of ICU admission of excluded patients? I think these are important to interpret the influence of selection bias on your results.
Response: We thank the reviewer for the important comment. COVID-19 patients with acute respiratory failure requiring high-flow nasal oxygen therapy, non-invasive ventilation, or invasive mechanical ventilation were eligible to be admitted to our ICU; thus, the inclusion criteria of acute respiratory failure were the need for high-flow oxygen therapy or mechanical ventilation. We added this information to the main manuscript, page 2, line 78. The reasons for ICU admission in excluded patients were any admission for surgery, stroke, MI, etc., but with no signs of acute respiratory failure and clear lungs on the chest x-ray. We hope that we have addressed the reviewer’s concern appropriately.
Reviewer: 2) How many patients missed the premorbid creatinine value? In some patients, the creatinine on the day of hospital admission might be higher than the premorbid one because of the AKI. If my understanding is correct, your study underestimated the AKI prevalence, because you applied the higher baseline creatinine value in some patients. Please confirm the correctness of your following sentence in discussion section. Page 11, Line 342-344 "Third, in many cases, we did not have a baseline creatinine value, and using the admission value may have overestimated the incidence of AKI."
Response: We thank the reviewer for this important comment. In around 60% of our patients, we did not have baseline creatinine before the ICU admission. We think that our sentence "Third, in many cases, we did not have a baseline creatinine value, and using the admission value may have overestimated the incidence of AKI" is correct. If you do not know the pre-hospitalization creatinine level and take the elevated ICU admission creatinine level as the baseline value, you might overestimate the AKI incidence since the pre-hospitalization creatinine value (true baseline) might already be elevated in case of unknown chronic kidney disease. Thus, you classify the patient as having AKI whereas, he does not have AKI, but chronic kidney disease. However, in our study, we think that this had a little impact on the AKI incidence as our population was young. We hope that we have addressed the reviewer's concern appropriately.
We thank the reviewer for his/her comments that have helped improve the manuscript's quality.
Reviewer 3 Report
This is a well-written one-center retrospective study. The study provides specific information on acute kidney injury in critically ill patients, an aspect about which little is still known. A strong point of the study is the analysis of inflammation markers such as IL6 in relation to AKI.
- It would be interesting to know the percentage of patients with elevated creatinine on admission for whom no previous creatinine was available.
- Is any treatment associated with a lower requirement of renal replacement therapy?
- Average duration and type of renal replacement therapy?
- 50% of patients with AKIN 2-3 required RRT. Was the RRT requirement included in the mortality models? If not, it should be explored.
Minor revisions:
- The IQR is a number, the range between the 25-75 quartile.
Author Response
Reviewer: This is a well-written one-center retrospective study. The study provides specific information on acute kidney injury in critically ill patients, an aspect about which little is still known. A strong point of the study is the analysis of inflammation markers such as IL6 in relation to AKI.
Response: We thank the reviewer for the supportive comment.
Reviewer: - It would be interesting to know the percentage of patients with elevated creatinine on admission for whom no previous creatinine was available.
Response: We thank the reviewer for this important point. Unfortunately, in around 60% of our patients, we did not have the pre-hospitalization baseline creatinine value. We stated this as one of the limitations of our study that might have resulted in an overestimation of our severe AKI incidence. However, we think that this had a little impact on our findings since most of our patients were young with supposed normal renal function before they were infected with the SARS-CoV-2 virus. Also, our severe AKI incidence rate is lower than in most of the previously reported studies. We hope that we have addressed all the reviewer's concerns appropriately.
Reviewer: - Is any treatment associated with a lower requirement of renal replacement therapy?
Response: We thank the reviewer for this comment. Our study did not investigate the risk factors associated with renal replacement therapy but with overall severe AKI. That is why we cannot answer this question directly. However, we did not find any treatment that was associated with a lower severe AKI incidence. We hope that we have addressed all the reviewer's concerns appropriately.
Reviewer: - Average duration and type of renal replacement therapy?
Response: We thank the reviewer for this comment. Again, our study did not focus on renal replacement therapy but on overall severe AKI development. Thus, we cannot answer the question regarding the duration of RRT. Regarding the type of RRT, we usually start with continuous RRT and shift to intermittent hemodialysis when the patients are more stable. We hope that we have addressed all the reviewer's concerns appropriately.
Reviewer: - 50% of patients with AKIN 2-3 required RRT. Was the RRT requirement included in the mortality models? If not, it should be explored.
Response: We thank the reviewer for this point. We did not include both RRT and severe AKI in the same model to avoid collinearity as the two variables are well correlated. We included only severe AKI since this was our aim. However, we replace severe AKI with RRT, the latter was also independently associated with mortality. Nevertheless, we did not show these results in the main text to avoid confusion as that was not part of our objectives. We hope that we have addressed all the reviewer's concerns appropriately.
Reviewer: Minor revisions: - The IQR is a number, the range between the 25-75 quartile.
Response: The reviewer is right. However, IQR can also be expressed as 25-75 quartile. In many studies, IQR was expressed as 25-75 quartile. We hope that we have addressed all the reviewer's concerns appropriately.
We thank the reviewer for his/her comments that have helped improve the manuscript's quality.